# A Scalable Reduced-Complexity Compression of Hyperspectral Remote Sensing Images Using Deep Learning

Sebastià Mijares i Verdú [1],*, Johannes Ballé [2], Valero Laparra [3], Joan Bartrina-Rapesta [1], Miguel Hernández-Cabronero [1] and Joan Serra-Sagristà [1]

1    Department of Information and Communications Engineering, Universitat Autònoma de Barcelona, 08193 Bellaterra, Spain; joan.bartrina@uab.cat (J.B.-R.); miguel.hernandez@uab.cat (M.H.-C.); joan.serra@uab.cat (J.S.-S.)

2    Google Research, Mountain View, CA 94043, USA; jballe@google.com

3    Image Processing Laboratory, Universitat de València, 46980 Paterna, Spain; valero.laparra@uv.es

\*    Correspondence: sebastia.mijares@uab.cat

**Abstract:** Two key hurdles to the adoption of Machine Learning (ML) techniques in hyperspectral data compression are computational complexity and scalability for large numbers of bands. These are due to the limited computing capacity available in remote sensing platforms and the high computational cost of compression algorithms for hyperspectral data, especially when the number of bands is large. To address these issues, a channel clusterisation strategy is proposed, which reduces the computational demands of learned compression methods for real scenarios and is scalable for different sources of data with varying numbers of bands. The proposed method is compatible with an embedded implementation for state-of-the-art on board hardware, a first for a ML hyperspectral data compression method. In terms of coding performance, our proposal surpasses established lossy methods such as JPEG 2000 preceded by a spectral Karhunen-Loève Transform (KLT), in clusters of 3 to 7 bands, achieving a PSNR improvement of, on average, 9 dB for AVIRIS and 3 dB for Hyperion images.

**Keywords:** image compression; hyperspectral; deep learning; data compression; AVIRIS; Hyperion

## 1. Introduction

Hyperspectral remote sensing scenes, either captured from aeroplanes such as by the Airborne Visible/Infrared Imaging Spectrometer (AVIRIS) or from space such as by the Hyperion sensor on board the EO-1 satellite, are of great interest for managing earth resources, natural disasters, or urban planning, among other applications [1]. According to the Union of Concerned Scientists, 42 satellites launched in the last decade have been put in orbit for the purpose of multispectral or hyperspectral Earth observation, as of May 2022 [2]. A growing number of hyperspectral images, such as those mentioned earlier, are being captured and stored, as demand for this kind of data continues to increase in the New Space era, which raises the pressure to develop novel compression techniques to further reduce data volume in transmission and long-term storage. Indeed, hyperspectral data compression is a particularly salient challenge in remote sensing, since for every spatial pixel sensed, numerous spectral measurements are taken and stored (one for every band). Due to the limited transmission capabilities of satellites and other remote sensing platforms, although vast amounts of hyperspectral data could be captured, not all of it can be transmitted down to Earth, limiting the amount of data sensed overall. For example, the HyspIRI sensor developed by NASA can produce up to 5 TB of data per day, but downlink capacity is limited [3]. When the bitrate constraints are especially tight, as they most often are for hyperspectral sensors, lossy compression is considered, in which case reconstructions must be of the highest possible quality, so as to not harm the information contained in these images.

In the last six years, the use of Machine Learning (ML) has produced a breakthrough in lossy compression for natural images [4–9], surpassing techniques such as JPEG [10], JPEG 2000 [11], and intra-frame HEVC [12]. ML compression has also been applied to remote sensing data [13–21]. These contributions have employed models presented in [4,6] as a baseline architecture, and are focused on different types of data. For example, some authors adapted the ML architectures to compress monoband images with bit-depth higher than 8 bits [13,14,20], others expanded on the baseline architectures for multispectral sensors [15–18], and some applied them in compression of hyperspectral data [19,21].

The state of the art of ML lossy compression for remote sensing can be organised according to whether the architecture exploits only 2D (spatial) redundancy, or both spatial and spectral redundancy. Regarding architectures that only consider 2D data, Alves de Oliveira et al. showed in [13] that applying the architecture proposed in [6] in a band-by-band fashion outperforms JPEG 2000 [11] for lossy compression. Other works published on ML compression of single-band remote sensing images include variants of the baseline architecture for SAR data, which modify the main transform with layers other than plain convolutions, such as residual blocks [22], or with a different structure altogether (pyramidal structure) [23]. Regarding contributions that use both 1D and 2D information during the learning stage, architectures for that purpose have been proposed using images from Landsat 8 and World-View 3 sensors in lossy regimes [24], compressed sensing with a learned decoder approaches [25], and codecs based on convolutional neural networks for volumetric compression AVIRIS images [19,21]. In general, due to computational restrictions on board, a volumetric ML compression method such as [19,21,24] for hyperspectral imaging is not practical in terms of computational cost. Furthermore, since the number of parameters in these networks needs to be expanded to compress images with larger numbers of bands, these methods' complexity increases more than linearly on the number of bands; in other words, they are not scalable on the number of bands. These restrictions are shown in detail in Section 3.

The practical adoption of ML techniques for hyperspectral data compression presents two key challenges: exploiting spectral correlation without incurring a too high computational cost, and scalability of the architecture in the number of spectral bands captured by different sensors. This second challenge is to make a design with the most linear computational complexity on the number of bands, so that it may remain practical for hyperspectral data with a larger number of bands. As explained, several works have been published on compression of higher bit-depth images exploiting 2D redundancy, and ML compression architectures for specific sensors with a fixed number of input bands have been proposed. This kind of proposal often incurs a high computational cost for on-board deployment. To address these challenges, in this paper, we propose compression in clusters of bands as a first step in the field of practical ML hyperspectral lossy data compression, as the complexity of the neural networks used increases quadratically with the number of input bands. This proposal is based on the Ballé et al. architecture [6], which is applied in clusters of three bands instead of on the entire volume at once, so as to reduce the computational cost of the algorithm. An additional benefit is the need to decompress a reduced number of bands when retrieving a specific band. Furthermore, in this proposal, the normalisation stage is modified to better encode and reconstruct data acquired at different wavelengths. This contribution surpasses the spectral KLT, followed by the JPEG 2000 coding scheme [26], when the KLT is also applied in clusters of up to 7 bands. The contributions presented in this paper are:

1.  The compression of an image in clusters of bands is studied as a scalable reduced-complexity compression method;
2.  A novel normalisation technique (range-adaptive normalisation) is proposed to avoid checkerboard effects in low-variance high bit depth images;
3.  Using a variant of an established neural compression network, the method is evaluated on two hyperspectral data sources, showing it is competitive at a cluster size compatible with off-the-shelf on-board hardware.

The rest of the paper is structured as follows. Section 2 introduces the end-to-end optimised transform coding paradigm this work is based upon. Section 3 describes the proposed method and the ML architectures used. Section 4 reports the conducted experiments. Finally, Section 5 provides our conclusions.

## 2. End-to-End Optimised Transform Coding

A successful approach for lossy image compression based on ML is end-to-end optimised transform coding, a paradigm in which the image $x$ is encoded by transforming it to a latent domain $y$ that is quantised into $\hat{y}$, and entropy encoded, producing a bitstream. Then, the bitstream is entropy decoded, obtaining $\hat{y}$, and transformed back to the original image domain, producing $\hat{x}$. Two neural networks act as the encoder, $g_a(\cdot, \theta_a)$, and decoder, $g_s(\cdot, \theta_s)$, transforms, where $\theta_a$ and $\theta_s$ are the parameters (weights) of these neural networks. The two transforms are jointly trained to minimise compression rate and distortion between the original and reconstructed images, hence the paradigm's name *end-to-end optimised transform coding* [7].

This is an autoencoder in a *rate-distortion optimisation* problem, where the rate is the expected code length (bitrate) of the compressed representation, and distortion is the difference between the original image $x$ and its reconstruction $\hat{x}$ under some metric, typically mean squared error (MSE). To perform entropy coding and calculate the bitrate in training, a prior probability model of the quantised representation is used, known to both encoder and decoder, namely the *entropy model* $p_{\hat{y}}$. Assuming the entropy coding technique is operating efficiently, the rate $R$ can be written as a cross entropy,

$$R = \mathbb{E}_{x \sim p_x} \big[ -\log_2 p_{\hat{y}}(Q(g_a(x, \theta_a))) \big], \tag{1}$$

where $Q$ represents the quantisation function, typically rounding to the nearest integer [6].

The entropy model can be a fixed probability distribution, as in [4], or it can be parameterised to adjust its probability estimate to each vector to be encoded, writing that entropy model as $p_{\hat{y}, \sigma}(\cdot)$. This parameterisation $\sigma$ can be computed from $\hat{y}$ using another neural network, a *hyperprior*. Since $\hat{y}$ is needed to produce the parameterisation $\sigma$, the auxiliary neural network is used both to calculate $\sigma$ and to be encoded. In other words, after the main transform has produced $y$, $y$ is further transformed into $z$ using a *hyper-analysis transform*, $h_a(\cdot, \phi_a')$, and then quantised into $\hat{z} = Q(z)$ to be transmitted as side information. The side information $\hat{z}$ is then decoded using a *hyper-synthesis transform*, $h_s(\cdot, \phi_s')$, into $\sigma$, which is the parameterisation used in the entropy model to encode $\hat{y}$. Together, these two later transforms conform the hyperprior. The rate of this side information can be calculated as in (1), obtaining the *loss function* of our end-to-end optimised codec,

$$\begin{aligned} L(x, \theta, \phi) = \mathbb{E}_{x \sim p_x} \big[ &-\log_2 p_{\hat{y}, \sigma}(Q(g_a(x, \theta_a))) \\ &-\log_2 p_{\hat{z}}(Q(h_a(g_a(x, \theta_a), \phi_a'))) \\ &+\lambda D(x, g_s(Q(g_a(x, \theta_a)), \theta_s)) \big], \end{aligned} \tag{2}$$

where $\lambda$ is a parameter to regulate the rate-distortion trade-off and $D(x, \hat{x})$ is the distortion function. In the case of the Ballé et al. architecture, the prior distribution is a multivariate Gaussian distribution with parametric scale $\sigma$ [6].

The gradient of $Q(\cdot)$ is zero almost everywhere; thus, in order to use gradient descent for optimisation, the problem needs to be relaxed. A common approach is to use additive uniform noise to replace the quantisation function; thus, in training, $Q$ is defined as

$$Q(x) = x + \varepsilon \text{ such that } \varepsilon \sim U\left(-\frac{1}{2}, \frac{1}{2}\right), \tag{3}$$

where $U\left(-\frac{1}{2}, \frac{1}{2}\right)$ is a multivariate uniform probability distribution. This rate-distortion optimisation problem is formally represented as a *variational autoencoder* (VAE) [27], where the encoder is understood to be an inference model and the decoder is read as a probabilistic generative model. Further details can be found in [6].

## 3. Proposed Method

The proposed method is based on the Ballé et al. architecture [6] shown in Figure 1. In that figure, blocks labelled "Conv $N \times k \times k/s$" indicate convolution with $N$ filters using $k \times k$ kernels with a stride of length $s$, and the arrow indicates downsampling or upsampling. General Divisive Normalisation (GDN) and Rectified Linear Units (ReLU) are used as activation functions. This proposal modifies the baseline in two aspects: one layer is removed from the hyperprior network, and the normalisation layer is modified. Inspired by the work of Alves de Oliveira et al. [13], our networks have a different number of filters in the hidden layers ($N$) and in the latent space ($M$). This allows us to increase the size of the latent space relative to the input space, while greatly reducing the computational cost of the network with respect to using the sample architecture applied to the entire spectrum. This architecture is used to compress hyperspectral images in clusters of three bands at a time, which is less computationally costly than feeding all the bands at once. The motivation for the computational cost reduction stems from the need to increase the number of filters in the network in proportion to the number of input image bands; this arises from the increased number of features to extract from the data (larger $N$) as well as to keep the ratio between the number of input samples and the number of output samples at a reasonable level (larger $M$). The complexity reduction resulting from this strategy is detailed in Section 3.2.

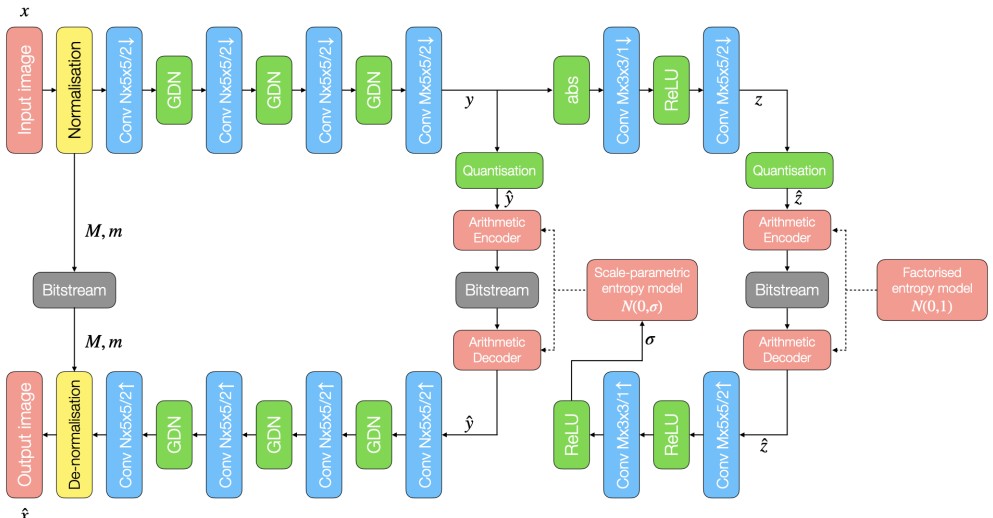

**Figure 1.** Architecture from [6] with variable normalisation.

### 3.1. Range-Adaptive Normalisation Layer

In ML image processing, data are normalised between 0 and 1 before being fed to the neural network to avoid issues such as exploding gradients in backpropagation [28]. In current ML compression architectures, 8 bit-depth images are normalised by dividing them by the dynamic range—from now on *uniform normalisation*—as

$$x'_{b,j,i} = \frac{x_{b,j,i}}{2^B - 1},\tag{4}$$

where $x$ and $x'$ are the input image and the normalised data, respectively. Indexes $b$, $j$ and $i$ denote the number of bands, rows and columns, and $B$ the bit-depth of the sensor. However, for 16 bit hyperspectral images, bands contain data acquired by the sensor at

different wavelengths, which is translated in data with different magnitude ranges that may not, in general, span the whole nominal range. Thus, employing (4) for 16 bit hyperspectral scenes with low variance will produce skewed normalised data.

When 16-bit images are decompressed using convolutional autoencoders, they tend to produce checkerboard artefacts, especially in low-variance data. This effect is produced because of the way the images are reconstructed using transposed convolution [29]. A similar artefact has been addressed using sub-pixel convolution [30] or resize-convolution [29]. However, in this context, these solutions may also produce checkerboard artefacts in low-variance data. Our proposed solution is *range-adaptive normalisation* instead of using (4), which is calculated as

$$x'_{b,j,i} = \frac{x_{b,j,i} - \text{MIN}_b}{\text{MAX}_b - \text{MIN}_b + 1}.$$

(5)

Range-adaptive normalisation consists of normalising every band $b$ independently, according on the minimum and maximum sample values in said $b$th band, denoted as $\text{MIN}_b$ and $\text{MAX}_b$, respectively. Note that $\text{MIN}_b$ and $\text{MAX}_b$ must be stored for each of the $k$ bands as side information. This side information is two 16-bit values per input band, a negligible amount overall. In the denominator, 1 is added for the special case of $\text{MIN}_b = \text{MAX}_b$ to avoid division by 0.

The capability of range-adaptive normalisation in removing the aforementioned artefact is evaluated using a factorised-prior model [4]. Figure 2 depicts the rate-distortion results comparing JPEG 2000, and [4] with uniform normalisation, sub-pixel convolution, and range-adaptive normalisation for Landsat 8 and AVIRIS images. Results indicate that range-adaptive normalisation performs on-par with sub-pixel convolution, especially in images such as AVIRIS, where ranges do not fully exploit the full dynamic range. Although sub-pixel convolution achieves good performance in lossy compression, it continues to produce the checkerboard artefact in low-variance bands—bands whose samples span a narrow range relative to the dynamic range—while the proposed range-adaptive normalisation variant does not, as Figure 3 shows. The number of bands with these characteristics depends on the sensor. For example, in AVIRIS data, there are around 50 or more such bands. Such bands can contain useful information, despite spanning a narrow section of the dynamic range, and due to their low variance, there is less tolerance to error in these samples.

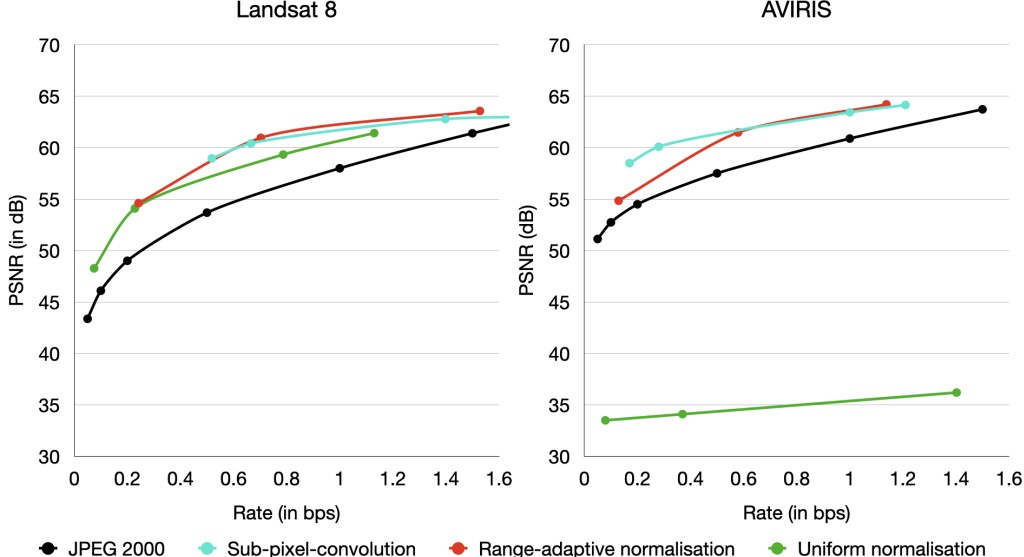

**Figure 2.** PSNR results of models trained and tested for band-by-band compression of Landsat 8 OLI and AVIRIS images.

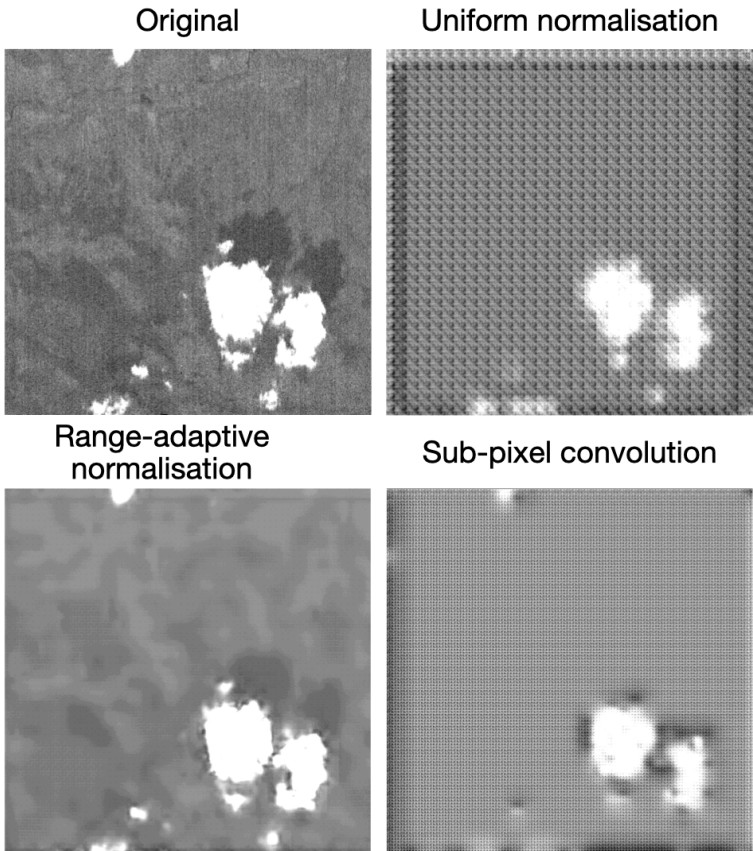

**Figure 3.** Crop of AVIRIS scene f090710t01p00r09rdn-b band 110, compressed at 0.05 bits per sample (bps).

### 3.2. Complexity Reduction

The computational complexity of the proposed method may be calculated by counting the number of operations per spatial pixel performed by our neural network, as in [13], considering $k$ input bands and $N$ and $M$ as the hidden and latent numbers of filters from the architecture in Figure 1. In our experiments for three-band images and work by other authors [13] on single-band images, the numbers of filters used follows the linear relation $N = 32(k+1)$ and $M = 128k$, which can be extrapolated to estimate the size of the network needed to compress clusters of $k$ bands.

Table 1 details such a calculation. There, *filters in* and *filters out* stand for the number of bands the input and output of each layer has. The number of kernels in a layer is the product of those two numbers. The size of those kernels is either $5 \times 5$ or $3 \times 3$, as indicated in Figure 1. The number of parameters of a convolutional layer is then calculated as

$$N_{params} = N_{in} \times N_{out} \times K^2, \tag{6}$$

where $K$ is the size of the kernel. The number of parameters in GDN layers is calculated identically to (6), adding $N_{out}$. Finally, the FLOPs/pixel is calculated as

$$\text{FLOPs/pixel} = \frac{N_{params}}{S^2}, \tag{7}$$

where $S$ is the stride accumulated until that layer in each spatial direction, so $S = 2$ in the first convolutional and GDN layers, $S = 4$ in the second convolutional and GDN layers, etc. This stride length is then reversed in the transposed convolution (TConv) and inverse GDN (iGDN) layers.

**Table 1.** Detailed complexity of the proposed architecture for $k$ input bands.

| Layer | Filters In | Filters Out | Parameters | FLOPs/Pixel |
|---|---|---|---|---|
| Norm. | $k$ | $k$ | 0 | $k$ |
| *Conv* | $k$ | $32(k+1)$ | $800k(k+1)$ | $200k(k+1)$ |
| *GDN* | $32(k+1)$ | $32(k+1)$ | $1024(k+1)^2 + 32$ | $256(k+1)^2 + 8(k+1)$ |
| *Conv* | $32(k+1)$ | $32(k+1)$ | $25{,}600(k+1)^2$ | $1600(k+1)^2$ |
| *GDN* | $32(k+1)$ | $32(k+1)$ | $1024(k+1)^2 + 32$ | $64(k+1)^2 + 2(k+1)$ |
| *Conv* | $32(k+1)$ | $32(k+1)$ | $25{,}600(k+1)^2$ | $400(k+1)^2$ |
| *GDN* | $32(k+1)$ | $32(k+1)$ | $1024(k+1)^2 + 32$ | $16(k+1)^2 + \frac{(k+1)}{2}$ |
| *Conv* | $32(k+1)$ | $128k$ | $102{,}400k(k+1)$ | $400k(k+1)$ |
| *HConv* | $128k$ | $128k$ | $147{,}456k^2$ | $576k^2$ |
| *HConv* | $128k$ | $128k$ | $409{,}600k^2$ | $400k^2$ |
| *THConv* | $128k$ | $128k$ | $409{,}600k^2$ | $400k^2$ |
| *THConv* | $128k$ | $128k$ | $147{,}456k^2$ | $576k^2$ |
| *TConv* | $128k$ | $32(k+1)$ | $102{,}400k(k+1)$ | $400k(k+1)$ |
| *iGDN* | $32(k+1)$ | $32(k+1)$ | $1024(k+1)^2 + 32$ | $16(k+1)^2 + \frac{(k+1)}{2}$ |
| *TConv* | $32(k+1)$ | $32(k+1)$ | $25{,}600(k+1)^2$ | $400(k+1)^2$ |
| *iGDN* | $32(k+1)$ | $32(k+1)$ | $1024(k+1)^2 + 32$ | $64(k+1)^2 + 2(k+1)$ |
| *TConv* | $32(k+1)$ | $32(k+1)$ | $25{,}600(k+1)^2$ | $1600(k+1)^2$ |
| *iGDN* | $32(k+1)$ | $32(k+1)$ | $1024(k+1)^2 + 32$ | $256(k+1)^2 + 8(k+1)$ |
| *TConv* | $32(k+1)$ | $k$ | $800k(k+1)$ | $200k(k+1)$ |
| Denorm. | $k$ | $k$ | 0 | $k$ |

Table 2 shows the total addition of the number of parameters and FLOPs/pixel. Due to the need to increase the number of filters in the network in proportion to the number of input bands, the complexity increases quadratically with respect to the number of input bands. A larger number of input bands means compressing fewer clusters. Given the number of bands in the image, $n$, our network encodes $\lceil \frac{n}{k} \rceil$ clusters. As a result, the overall number of operations per spatial pixel by the encoder is as in Equation (8), which is clearly monotonically increasing with respect to $k$. Thus, using larger clusters results in greater computational cost. The number of floating-point operations (FLOPs) per sample—operations per pixel per band—is equal to those per pixel divided by the number of bands, $n$.

$$\lceil \frac{n}{k} \rceil (3912k^2 + 5283k + 2348) \text{ FLOPs/pixel} \approx 3912kn + 5283n + 2348\frac{n}{k} \text{ FLOPs/pixel} \quad (8)$$

Considering images with 224 bands compressed using a single 224-band cluster (AVIRIS) requires 881,581 FLOPs/sample, while using three-band clusters would require 17,802 FLOPs/sample to encode, thus the proposed method provides a 98% reduction in complexity in this particular case. On the other hand, the Mijares et al. 2021 method [31] for AVIRIS data in four-band clusters requires 18,097 FLOPs/sample, more complex than the proposed method. For completeness sake, we also mention that this complexity is two orders of magnitude higher than that of the CCSDS 122.1-B-1 or JPEG 2000 standards [32,33]. This number of encoder operations of the proposed method (three-band clusters) is compatible with an embedded implementation on board using hardware, such as the Movidius Myriad 2 from Intel [13], and, by extension, with more capable and efficient state-of-the-art hardware.

**Table 2.** Addition of complexity of the proposed architecture for *k* input bands.

| | Parameters | FLOPs/Pixel |
|---|---|---|
| **Total** | $1.4 \cdot 10^6 \, k^2 + 4.2 \cdot 10^5 \, k + 10^5$ | $7.8 \cdot 10^3 k^2 + 1.1 \cdot 10^4 \, k + 4.7 \cdot 10^3$ |
| **Encoder** | $7.1 \cdot 10^5 \, k^2 + 2.1 \cdot 10^5 \, k + 5.4 \cdot 10^4$ | $3.9 \cdot 10^3 \, k^2 + 5.3 \cdot 10^3 \, k + 2.3 \cdot 10^3$ |

The complexity of the proposed method can be compared to that in [19]; when also applied to 224-band AVIRIS images, we would have the results in Table 3. Note that we have made an assumption on the size of the convolutional kernels as being $5 \times 5$, since that is not explicitly mentioned in their paper. Observe that their convolutions are applied before downsampling (which is performed with a Max Pool layer), which makes them costlier. This complexity translates to 3445 FLOPs/sample. Indeed, this appears much lower than our proposed method, however the dimensionality reduction performed by this transform is 448:1 which, if we use 32-bit values, would give us a maximum bitrate of 0.07 bps, not considering any additional reduction by an arithmetic coder as is used. To compress at higher bitrates, as needed in practice, this architecture needs to be scaled up in the number of filters used. To compress at an absolute maximum of 1 bps, this scaling has to be $\times 14$, which would yield the complexity in Table 4. That corresponds to 92,415 FLOPs/sample, much larger than we can use.

**Table 3.** Detailed complexity of the Dua et al. 2021 [19] architecture for 224-band data.

| Layer | Filters In | Filters Out | Parameters | FLOPs/Pixel |
|---|---|---|---|---|
| Norm. | 224 | 224 | 0 | 224 |
| *Conv* | 224 | 128 | 716,800 | 716,800 |
| *ReLU* | 128 | 128 | 0 | 128 |
| *Conv* | 128 | 64 | 204,800 | 51,200 |
| *ReLU* | 64 | 64 | 0 | 64 |
| *Conv* | 64 | 32 | 51,200 | 3200 |
| *TanH* | 32 | 32 | 0 | 32 |
| *TConv* | 32 | 64 | 51,200 | 3200 |
| *ReLU* | 64 | 64 | 0 | 64 |
| *TConv* | 64 | 64 | 102,400 | 6400 |
| *ReLU* | 64 | 64 | 0 | 64 |
| *TConv* | 64 | 128 | 204,800 | 51,200 |
| *ReLU* | 128 | 128 | 0 | 128 |
| *TConv* | 128 | 128 | 409,600 | 102,400 |
| *ReLU* | 128 | 128 | 0 | 128 |
| *TConv* | 128 | 224 | 716,800 | 716,800 |
| *ReLU* | 224 | 224 | 0 | 224 |
| *TConv* | 224 | 224 | 1,254,400 | 1,254,400 |
| *ReLU* | 224 | 224 | 0 | 224 |
| Denorm. | 224 | 224 | 0 | 224 |
| **Total** | | | **3,712,000** | **2,906,880** |
| **Encoder** | | | **972,800** | **771,648** |

**Table 4.** Detailed complexity of the Dua et al. 2021 [19] architecture for 224-band data, scaled ×14 for ≤1 bps compression.

| Layer | Filters In | Filters Out | Parameters | FLOPs/Pixel |
|:---:|:---:|:---:|:---:|:---:|
| Norm. | 224 | 224 | 0 | 224 |
| *Conv* | 224 | 1792 | 10,035,200 | 10,035,200 |
| *ReLU* | 1792 | 1792 | 0 | 1792 |
| *Conv* | 1792 | 896 | 40,140,800 | 10,035,200 |
| *ReLU* | 896 | 896 | 0 | 896 |
| *Conv* | 896 | 448 | 10,035,200 | 627,200 |
| *TanH* | 448 | 448 | 0 | 448 |
| *TConv* | 448 | 896 | 10,035,200 | 627,200 |
| *ReLU* | 896 | 896 | 0 | 896 |
| *TConv* | 896 | 896 | 20,070,400 | 1,254,400 |
| *ReLU* | 896 | 896 | 0 | 896 |
| *TConv* | 64 | 1792 | 40,140,800 | 10,035,200 |
| *ReLU* | 1792 | 1792 | 0 | 1792 |
| *TConv* | 1792 | 1792 | 80,281,600 | 20,070,400 |
| *ReLU* | 1792 | 1792 | 0 | 1792 |
| *TConv* | 1792 | 224 | 10,035,200 | 10,035,200 |
| *ReLU* | 224 | 224 | 0 | 224 |
| *TConv* | 224 | 224 | 1,254,400 | 1,254,400 |
| *ReLU* | 224 | 224 | 0 | 224 |
| Denorm. | 224 | 224 | 0 | 224 |
| **Total** | | | **222,028,800** | **63,983,584** |
| **Encoder** | | | **60,211,200** | **20,700,960** |

## 4. Experimental Results

In the field of remote sensing data compression, there are three main established lossy compression standards: JPEG 2000 [11], CCSDS 122.1-B-1 [32], and CCSDS 123.0-B-2 [34]. For hyperspectral data, using the Karhunen–Loève Transform (KLT) for spectral decorrelation in combination with a Discrete Wavelet Transform (DWT) for spatial decorrelation (such as in JPEG 2000) is in widespread use as well [35–37], where KLT + JPEG 2000 outperforms KLT + CCSDS 122.1-B-1 and CCSDS 123.0-B-3. Therefore, JPEG 2000 and KLT + JPEG 2000 are chosen as benchmarks for coding hyperspectral remote sensing images. Other neural compression of hyperspectral data proposals, such as [19], are of much higher computational cost than our method, or what can be assumed in onboard compression, and thus are not included in our comparison.

Two data sets from different sensors are used in our experiments. For the AVIRIS sensor, a collection of calibrated scenes [38] from a variety of sites across North America is used, consisting of 180 scenes for training and 20 scenes for testing. All of the scenes are 512 × 512 pixels with 224 spectral bands, for a total of 21 GB of training data. For the Hyperion sensor deployed in the EO-1 NASA mission, 71 scenes are used for training and 13 for testing, obtained from the United States Geological Survey (USGS) Earth Explorer [39]. The pre-processing described in [26] is applied to these images to eliminate line artefacts [40]. Scenes from this sensor are 1024 × 256 pixels and have 242 spectral bands, adding to a total of 9 GB of training data.

All the models tested were trained for no less than 15.000 iterations on each dataset using Adam [41] as our optimizer, with MSE as the distortion metric in the loss function. Since our proposed model is applied in clusters of three bands, in order to produce a fair comparison, the KLT is also applied in clusters of three bands. JPEG 2000 can either be applied to each of these clusters independently or to the full spectrally transformed volume. Since our models do not allocate bits across clusters, the first of those options makes a fair comparison. Our models can be found in a GitHub repository (https://github.com/smijares/mblbcs2023, accessed on 25 July 2023).

Figure 4 depicts the average compression Peak Signal-to-Noise Ratio (PSNR) results of full volumes in clusters of three bands for AVIRIS and Hyperion images. The vertical and horizontal axes provide the PSNR and the rate in bits per sample (bps), respectively. Similarly, Figure 5 depicts the average compression Relative Squared Error (RSE) results, which is another metric based on Mean Squared Error (MSE). The PSNR is calculated as $PSNR = 20 \log_{10} \left( \frac{2^{16}-1}{\sqrt{MSE}} \right)$, while the RSE is calculated as $RSE = \frac{MSE}{\frac{1}{N} \sum_{i=1}^{N} (x_i - \bar{x})^2}$, where $\bar{x}$ is the mean of the values in the image. These plots provide results for JPEG 2000, when the KLT is applied to clusters of three bands, and the transformed clusters are independently compressed with JPEG 2000 (3-band KLT + JPEG 2000) and, when the KLT is applied to clusters of three bands and the transformed, clusters are compressed with JPEG 2000 considering the whole volume (three-band KLT + Full volume JPEG 2000). For the proposed method, we set parameters $N = 128$ and $M = 384$. The proposed method achieves competitive results compared to KLT + JPEG 2000 applied in clusters of three bands in both types of data, surpassing it by 9 dB PSNR on average in AVIRIS data and by 3 dB PSNR in Hyperion images. Furthermore, the proposed method can match the performance of three-band KLT + JPEG 2000 applied to the full volume.

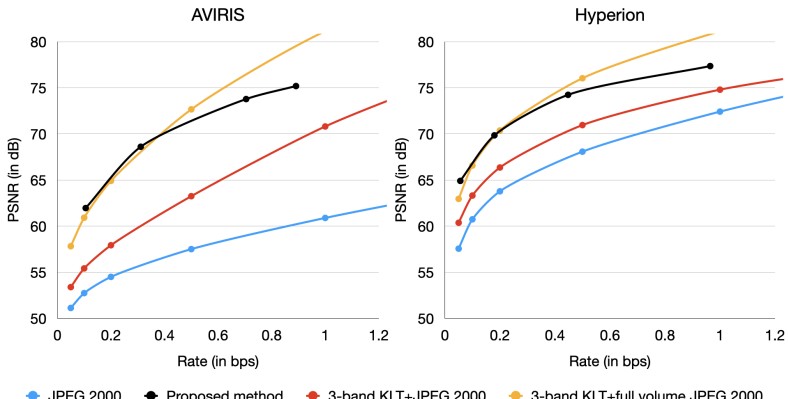

**Figure 4.** PSNR performance of models trained and tested for compression in clusters of 3 bands of AVIRIS and Hyperion images.

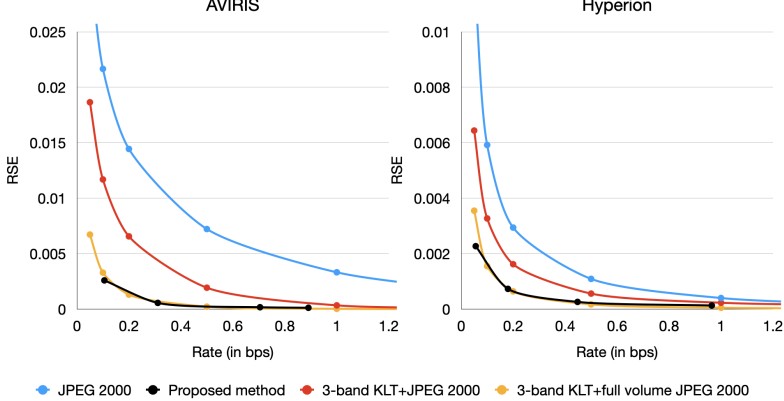

**Figure 5.** RSE performance of models trained and tested for compression in clusters of 3 bands of AVIRIS and Hyperion images.

Measuring the spectral angle (SA) of our reconstructions, the proposed method also outperforms KLT + JPEG 2000 in clusters of three bands. Figure 6 depicts SA at different rates for the same techniques and configurations as in Figure 4. Figures 7 and 8 show a visual comparison of different bands for the AVIRIS and Hyperion sensors, respectively. It is clear our models produce better reconstructions at higher-variance bands, where KLT + JPEG 2000 tends to blur certain details which our models kept (such as the river in the AVIRIS image), while at lower-variance bands, our models and KLT + JPEG 2000 perform more on par.

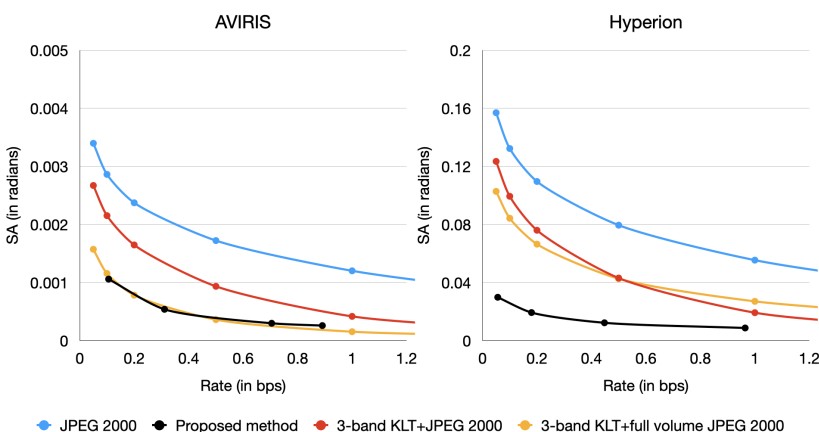

**Figure 6.** Spectral angle performance of models trained and tested for compression in clusters of 3 bands of AVIRIS and Hyperion images.

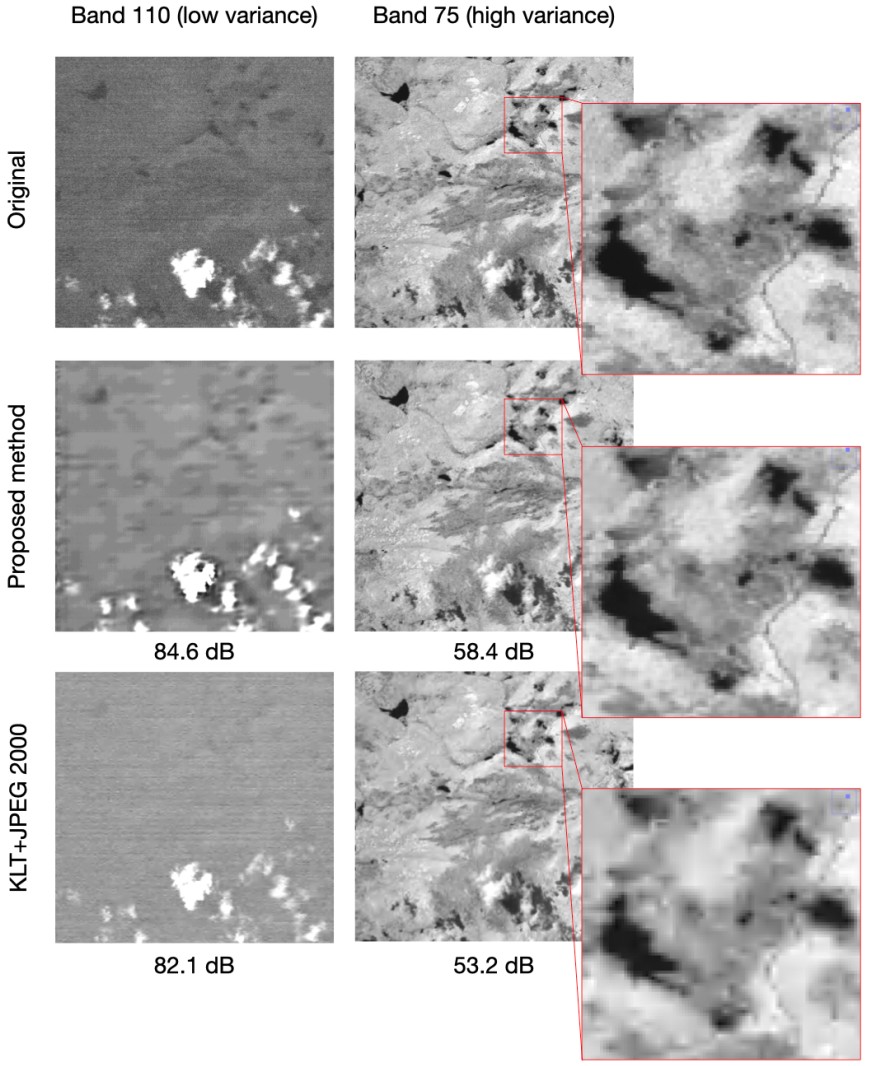

**Figure 7.** Reconstruction of some selected bands from AVIRIS scene f090710t01p00r09rdn-b using our method, with zoom-in to better appreciate some od the differences. The overall image is compressed at 0.11 bps and overall reconstruction PSNR is 60.92 dB.

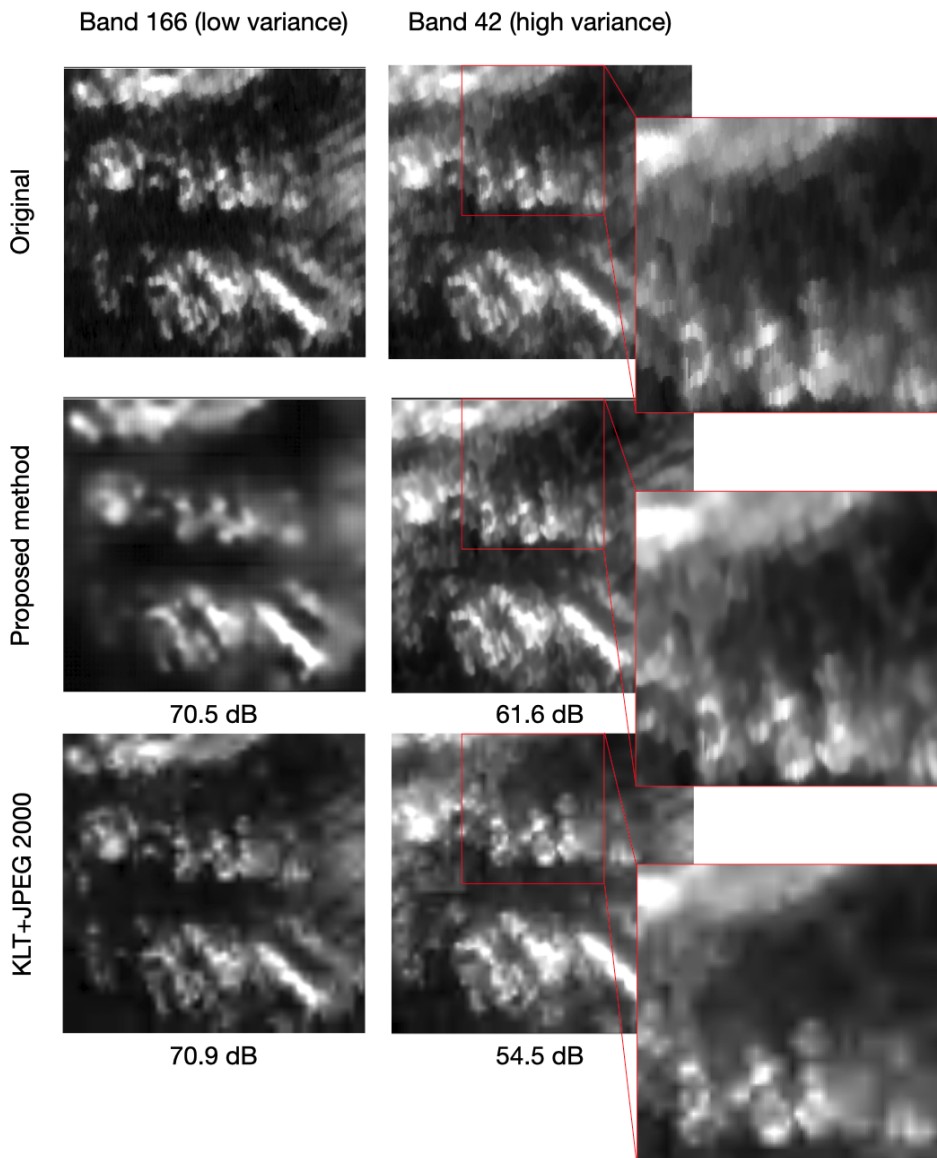

**Figure 8.** Reconstruction of some selected bands from Hyperion scene EO1H1980182017066110K3 using our method, with zoom-in to better appreciate some od the differences. The overall image was compressed at 0.05 bps and overall reconstruction loss was 63.78 dB PSNR.

The proposed method for compressing the images in clusters of three bands not only surpasses KLT + JPEG 2000 when applied on those same clusters, but can also perform on par with KLT + JPEG 2000 applied in clusters of 7 bands, and the Mijares et al. 2021 method [31] for AVIRIS data in four-band clusters, as shown in Table 5 for AVIRIS data and in Table 6 for Hyperion data.

**Table 5.** Average compression performance comparison on AVIRIS data.

| Compression Method | PSNR at 0.1 bps | PSNR at 0.3 bps |
| --- | --- | --- |
| JPEG 2000 | 52.74 dB | 55.41 dB |
| 3 bands KLT + JPEG 2000 | 55.42 dB | 59.54 dB |

**Table 5.** *Cont.*

| 5 bands KLT + JPEG 2000 | 57.01 dB | 62.57 dB |
|---|---|---|
| 7 bands KLT + JPEG 2000 | 61.26 dB | 67.66 dB |
| Mijares et al. 2021 [31] (4 bands) | 58.47 dB | 65.45 dB |
| Proposed method (3 bands) | **61.96 dB** | **68.61 dB** |

**Table 6.** Average compression performance comparison on Hyperion data.

| Compression Method | PSNR at 0.05 bps | PSNR at 0.45 bps |
|---|---|---|
| JPEG 2000 | 57.56 dB | 67.23 dB |
| 3 bands KLT + JPEG 2000 | 60.37 dB | 70.05 dB |
| 5 bands KLT + JPEG 2000 | 62.28 dB | 71.64 dB |
| 7 bands KLT + JPEG 2000 | 64.23 dB | 73.25 dB |
| Proposed method (3 bands) | **64.91 dB** | **74.24 dB** |

## 5. Conclusions

In this work, a novel method for the compression of hyperspectral remote sensing images based on neural networks and a clustering strategy is proposed. This codec is competitive with key conventional methods such as KLT + JPEG 2000 on AVIRIS and Hyperion data. The proposal of compression in clusters of bands is scalable to images with any number of bands, as opposed to using these architectures with an arbitrary number of input bands, which is impractical as the number of parameters in the network would be scaled accordingly. Such a clustering approach is a first step in practical compression techniques to be deployed in scenarios with limited resources, such as on-board satellites. The proposed method in this paper could be implemented to run in state-of-the-art on-board hardware at the present date. Furthermore, range-adaptive normalisation alone is a highly effective method for enhancing the performance of convolutional neural networks in compression of low-variance 16-bit images, which in turn avoids checkerboard artefacts in low-variance images that other tested architectures produce. Results here presented serve as a starting point for lower-complexity designs that could be deployed on-board while remaining competitive with present in-use standards such as KLT + JPEG 2000.

**Author Contributions:** Conceptualisation, S.M.i.V., J.B., V.L. and J.S.-S.; methodology, S.M.i.V., J.B., V.L. and J.S.-S.; software, S.M.i.V. and J.B.; validation, J.B.-R. and M.H.-C.; formal analysis, S.M.i.V.; investigation, S.M.i.V.; resources, J.B.-R. and J.S.-S.; data curation, S.M.i.V.; writing—original draft preparation, S.M.i.V.; writing—review and editing, J.B., V.L., J.B.-R., M.H.-C. and J.S.-S.; visualisation, S.M.i.V.; supervision, J.B.-R. and J.S.-S.; project administration, J.S.-S. All authors have read and agreed to the published version of the manuscript.

**Funding:** This research was funded by the Spanish Ministry of Economy and Competitiveness and the European Regional Development Fund under grants RTI2018-095287-B-I00 (MINECO/FEDER, UE) and PID2020-118071GB-I00/AEI/10.13039/501100011033, the regional grant GV/2021/074, the Beatriu de Pinós programme 2018-BP-00008 and the Consolidated Research Group of Catalonia SGR2021-00643, funded by the Government of Catalonia, and the Horizon 2020 Marie Skłodowska-Curie agreement #801370.

**Data Availability Statement:** Our test data are sourced from public repositories and can be downloaded at gici.uab.cat.

**Conflicts of Interest:** The authors declare no conflict of interest.

## Abbreviations

The following abbreviations are used in this manuscript:

| | |
|---|---|
| AVIRIS | Airborne Visible/Infrared Imaging Spectrometer |
| KLT | Karhunen–Loève Transform |
| ML | Machine Learning |
| PSNR | Peak Signal-to-Noise Ratio |
| SA | Spectral Angle |
| FLOPs | Floating Point Operations |

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
