# Peer review of "A Scalable Reduced-Complexity Compression of Hyperspectral Remote Sensing Images Using Deep Learning"

_remotesensing, doi:10.3390/rs15184422_

Round 1

Reviewer 1 Report

In this paper, the authors propose a hyperspectral data compression method based on deep learning. The structure of this paper is clear; however, there are some issues that should be addressed as listed in detail as follows:

1. In Abstract, the authors states that, current hyperspectral compression methods are facing the problem ofcomputational complexity and scalability for large numbers of bands. Please clarify the limitations in detail.

2. This kind of proposals often incurs into high computational cost for on-board deployment. Please clarify it.

3. It is suggested to rewrite the Abstract. Please clarify the problem, and how you solve the problem and what the novelty of the paper is.

 4. Deep learning based hyperspectral compression models should be reviewed as well since these models are more popular at present. In addition, clarify the problems of these models in Introduction, and the advantage of the proposed model against these models.

5.  Many recently published works for hypersepctral compression, please refer to or compare with these works.

6. It is suggested to conclude the contributions of this paper in the Introduction Part.

Moderate editing of English language required.

Reviewer 2 Report

In the article titled Scalable Reduced-Complexity Compression of Hyperspectral Remote Sensing Images Using Deep Learning, author proposes a new method based on neural network clustering strategy, which achieves lower computational complexity and can be extended for different data sources with different frequency bands. It is also hardware compatible and practical, but there are still some issues in certain parts of the paper. Here are some questions and suggestions:

1、In the range-adaptive normalization and sub-pixel convolution method (line 157), you mentioned that the checkerboard artefact only occurs in certain situations. What is the frequency of these situations? Which method is more suitable in most cases?

2、In Figure 4-5, only traditional compression methods (such as JPEG2000 and CCDS123) are used for comparison. Are there any machine learning image compression methods in the same field for comparison in recent years?

3、It is suggested to add a relative quadratic error (RQE) comparison of the spectral curve to the result section.

Round 2

Reviewer 1 Report

I have no more comments.